# Genetic Profile of Left Ventricular Noncompaction Cardiomyopathy in Children—A Single Reference Center Experience

**DOI:** 10.3390/genes13081334

**Published:** 2022-07-26

**Authors:** Dorota Piekutowska-Abramczuk, Agata Paszkowska, Elżbieta Ciara, Kamila Frączak, Alicja Mirecka-Rola, Dorota Wicher, Agnieszka Pollak, Karolina Rutkowska, Jędrzej Sarnecki, Lidia Ziółkowska

**Affiliations:** 1Department of Medical Genetics, The Children’s Memorial Health Institute, 04-730 Warsaw, Poland; e.ciara@ipczd.pl (E.C.); k.fraczak@ipczd.pl (K.F.); d.wicher@ipczd.pl (D.W.); 2Department of Cardiology, The Children’s Memorial Health Institute, 04-730 Warsaw, Poland; a.paszkowska@ipczd.pl (A.P.); a.mirecka-rola@ipczd.pl (A.M.-R.); 3Department of Medical Genetics, Medical University of Warsaw, 02-106 Warsaw, Poland; poli25@wp.pl (A.P.); karolina.rutkowska901@gmail.com (K.R.); 4Department of Diagnostic Imaging, The Children’s Memorial Health Institute, 04-730 Warsaw, Poland; j.sarnecki@ipczd.pl

**Keywords:** left ventricular noncompaction, cardiomyopathy, next generation sequencing, molecular etiology, heart failure, arrhythmia, children

## Abstract

Background: Left ventricular noncompaction cardiomyopathy (LVNC) is a rare cardiac disorder characterised by the presence of a two-layer myocardium with prominent ventricular trabeculation, intertrabecular deep depressions and an increased risk of heart failure, atrial and ventricular arrhythmias and systemic thromboembolic events in affected patients. The heterogeneous molecular aetiology solved in 10%–50% of patients more frequently involves sarcomeric, cytoskeletal or ion channel protein dysfunction—mainly related to causative *MYH7*, *TTN* or *MYBPC3* variants. The aim of the study was to determine the molecular spectrum of isolated LVNC in a group of children examined in a single paediatric reference centre. Methods: Thirty-one paediatric patients prospectively diagnosed with LVNC by echocardiography and cardiovascular magnetic resonance examination were recruited into the study group. The molecular analysis included next-generation sequencing (gene panel or whole exome) and classic Sanger sequencing. All selected variants with high priority were co-segregated in the available parents. Results: We identified 16 distinct variants in 11 genes in 16 patients (52%), including 10 novel alterations. The most frequent defects in our cohort were found in the genes *HCN4* (*n* = 4), *MYH7* (*n* = 2) and *PRDM16* (*n* = 2). Other likely disease-causing variants were detected in *ACTC1, ACTN2, HCCS, LAMA4, MYH6, RBM20, TAFFAZIN* and *TTN*. Patients with established molecular defects more often presented with arrhythmia, thromboembolic events and death, whereas the predominant symptoms in patients with no identified molecular defects were heart failure and the presence of late gadolinium enhancement. Conclusion: This study expands the genetic and clinical spectrum of childhood LVNC. Although the molecular aetiology of LVNC varies widely, the comprehensive testing of a wide panel of cardiomyopathy-related genes helped to identify underlying molecular defects in more than half of the children in the study group. The molecular spectrum in our cohort correlated with the occurrence of arrhythmia, death and a family history of cardiomyopathy. We confirmed that genetic testing is an integral part of the work-up and management LVNC in children.

## 1. Introduction

Left ventricular noncompaction cardiomyopathy (LVNC) is a rare cardiac disorder characterised by the presence of a two-layer myocardium and comprising compacted and noncompacted segments of the left ventricular (LV) muscle with prominent ventricular trabeculation and intertrabecular deep depressions communicating with the LV cavity [1]. LVNC either occurs in the isolated form (i-LVNC) or is associated with congenital heart disease [1,2,3]. Mixed phenotypes of LVNC and other cardiomyopathies (CMP)—including dilated (LVNC-DCM), hypertrophic (LVNC-HCM), restrictive (LVNC-RCM) or arrhythmogenic (LVNC-ACM) forms—are frequently reported [1,4]. Patients with LVNC are at increased risk of heart failure (HF), atrial and ventricular arrhythmias and systemic thromboembolic events [5].

The increasing knowledge of CMP pathogenesis has resulted in a higher recognition rate of LVNC in routine clinical practice [1]. The basic diagnostic imaging study workflow includes echocardiography (ECHO), utilising the most commonly accepted criteria proposed by Jenni et al. [6], and cardiovascular magnetic resonance imaging (CMR) with the Petersen criteria [7]. 

The mechanisms leading to the LVNC phenotype mostly remain unclear, though some authors have pointed to an abnormal cardiac embryogenesis resulting from the intrauterine arrest of normal ventricular myocardium maturation and compaction [8]. Recent findings from Wu et al. [9] indicated the improper transcriptional specification of compact or trabecular cardiomyocytes as a potential common mechanism in LVNC development. 

In most cases, the knowledge about the genetics of LVNC is based on the results of studies performed on adult patients. Generally, it was stated that familial LVNC cases accounted for 20–40% of patients [1], while genetic inheritance could be molecularly confirmed in 10–50% of patients in studies of different sizes, and was detected more frequently in children than in adults [10]. However, there are limited and inconsistent data regarding genetic test results in paediatric patients, including the molecular spectrum and diagnostic yield. LVNC is a genetically heterogeneous disease with multiple genes involved [11]. The majority of causative molecular variants occur in the genes that encode sarcomeric, cytoskeletal or ion channel proteins, with *MYH7*, *TTN* and *MYBPC3* being the most often affected [10]. Additionally, the metabolic/mitochondrial, NOTCH signalling pathway or large chromosomal abnormalities are involved in LVNC aetiology in children [1]. The first genetic dysfunction reported in children with LVNC was the X-linked *TAFAZZIN* gene, which is associated with Barth syndrome [12]. Despite these insights, defining specific genotype–phenotype correlations remains a challenge [1].

The purpose of our study was to determine the molecular spectrum of an isolated form of LVNC in a group of children examined in a single paediatric reference centre, and to expand the knowledge of the genetic aetiology of this rare cardiomyopathy.

## 2. Materials and Methods

### 2.1. Patient Selection and Clinical Data Collection

Thirty-one paediatric patients under the age of 18 years who were hospitalised between February 2008 and December 2021 in the Department of Cardiology of the Children’s Memorial Health Institute (CMHI) with a diagnosis of isolated LVNC confirmed by echocardiography and CMR were included in the study. The exclusion criterion for the study was the coexistence of LVNC with congenital heart disease. The echocardiographic criteria for LVNC diagnosis were (1) the presence of a two-layer structure with a compacted (C) and noncompacted (NC) myocardial layer of trabecular meshwork with deep endomyocardial spaces, (2) a maximum end-systolic ratio of the NC/C layers of greater than 2 and (3) colour Doppler evidence of deep perfused intertrabecular recesses [6]. 

All patients underwent comprehensive clinical, electrocardiographic and echocardiographic tests, CMR imaging, laboratory analysis and genetic studies. The data also comprised family history and follow-up data where available. The clinical profile of some of these patients was presented in our previous publications [13,14]. The clinical complaints, NYHA/Ross functional class, enlargement and LV systolic dysfunction and N-terminal type B natriuretic propeptide (NTproBNP) values in blood serum were assessed in terms of symptoms of HF. 

The 12-lead resting ECG with an assessment of heart rate (HR), 24 h Holter electrocardiography (ECG) with an analysis of minimal, maximal and average HR and the occurrence of sinus pauses (RR pauses) of longer than 2 s were analysed and compared to the reference values in the literature [15,16]. Echocardiography imaging was performed using a Philips Epiq7 (Philips Medical Systems, Bothell, WA). The echocardiographic diagnosis of LVNC was based on Jenni’s criteria when the LV NC/C ratio was 2.0 or greater, measured in the parasternal short-axis view in the end-systolic phase and below the papillary muscle [6]. Echocardiographic measurements included LV end-diastolic diameter (LVEDd) and LV ejection fraction (LVEF), which were assessed using Simpson’s method. The results were referenced against the available paediatric normative values [17,18]. LV enlargement was diagnosed when the LVEDd z-score was greater than 2; LV systolic function impairment was defined as an LVEF of 55% or less. The methodology of the echocardiographic examination was described in detail in an earlier publication [19]. 

CMR studies were performed with a 1.5T magnetic resonance scanner (Magnetom AvantoFit, Siemens, Erlangen, Germany). The results were analysed on a dedicated diagnostic workstation using CVi42 software (Circle, Calgary, Canada). LVNC was diagnosed based on Petersen’s criteria [7] if the NC/C ratio at the end-diastole was greater than 2.3. Late gadolinium enhancement (LGE) images were obtained 10–15 min after the intravenous administration of 0.1 mmol/kg of gadobutrol (Gadovist, Bayer, Berlin, Germany) and were visually evaluated for areas of myocardial hyperintensity visible in two distinct planes. The extent of LGE was quantitatively assessed and determined as a percentage of the LV total mass, as described in our earlier study [19].

### 2.2. Molecular Studies and Data Analysis

DNA samples were automatically extracted from the peripheral blood of each participant with a MagCore Nucleic Acid Extractor HF16Plus (RBC Bioscience, New Taipei City, Taiwan) according to the manufacturer’s protocol. Next-generation sequencing (NGS) was performed using a HiSeq 1500 system (Illumina, San Diego, CA, USA) with the original CMHI NGS panel of 1,000 clinically relevant genes (Roche Diagnostics, Rotkreuz, Switzerland), and in one case with a TruSight One Sequencing Panel (Illumina, San Diego, CA, USA). In two cases, whole exome sequencing (WES) was applied as a subsequent molecular test. Copy number variations (CNVs) were identified with a CNV kit [20] using a reference consisting of 30 samples. 

NGS analysis was conducted using a set of known cardiomyopathy-associated genes (Appendix A) following the study protocol described previously [21]. After sequence alignment and variant calling—with standard filtering starting with the functional variant effect—only single nucleotide missense, nonsense and splice-site variants, insertions and deletions were selected for further analysis. When considering *TTN* alterations, only loss-of-function variants were evaluated. Any identified variants with a minor allele frequency of <0.005 were then filtered out as reported in the Genome Aggregation Database (gnomAD; https://gnomad.broadinstitute.org; accessed on 21 March 2022), the Exome Variant Server (EVS; https://evs.gs.washington.edu/EVS/; accessed on 21 March 2022), the UK10K Project (https://www.uk10k.org/; accessed on 21 March 2022) and an in-house database comprising more than 5,000 Polish individuals with unrelated diseases (Department of Medical Genetics [DMG]; accessed on 21 March 2022). Then, the variants were evaluated after considering their predicted impact on protein structure and function with in silico algorithms, including CADD, FATHMM, MetaLR, MetaSVM, LRT, MutationAssessor, MutationTaster, PolyPhen2 (HDIV and HVAR) and SIFT as well as MaxEnt, NNSPLICE and SSF (the last three algorithms for splicing variant assessment are incorporated into the software programme Alamut Visual^TM^ Plus (Interactive Biosoftware, Rouen, France; www.interactive-biosoftware.com)). Variants with pathogenicity indicated by 4 or more algorithms were considered for further evaluation. Finally, we carefully reviewed the literature—the Online Mendelian Inheritance in Man database (OMIM; https://www.ncbi.nlm.nih.gov/omim; accessed on 23 March 2022), ClinVar (https://www.ncbi.nlm.nih.gov/clinvar; accessed on 23 March 2022) and the Human Gene Mutation Database Professional 2021 v. 4 (HGMD; http://www.hgmd.cf.ac.uk; accessed on 23 March 2022) to confirm the clinical relevance and any genotype–phenotype correlations. The NGS reads were visualised using Integrated Genomic Viewer (IGV; https://software.broadinstitute.org/software/igv; accessed on 16 March 2022). All variants with high priority were confirmed by Sanger sequencing in the proband and segregated in available parents. For patient P10—suspected for Barth syndrome—an analysis directed at the *TAFAZZIN* gene (PCR and Sanger sequencing) was performed. The variants under study were classified according to the guidelines of the American College of Medical Genetics and Genomics and the Association of Molecular Pathology [22] as pathogenic (P), likely pathogenic (LP) and variant of uncertain significance (VUS). 

### 2.3. Ethics Statement

The CMHI Ethics Committee approved the protocol of this study and informed consent was obtained from all individual participants

## 3. Results

### 3.1. Clinical Characteristics of the Study Group

A total of 31 patients from 29 families (there were two sets of siblings: P2/P3 and P30/P31) diagnosed with isolated LVNC were included in the study. This group included 18 girls and 13 boys. The median age of the patients was 11 years (IQR: 6–13). 

In echocardiography, the median NC/C ratio in all 31 patients was 2.80 (IQR: 2.24–3.65), which met Jenni’s criteria for the diagnosis of LVNC. In 25 children, CMR imaging was performed to further assess LV morphology and function and to screen for features of myocardial fibrosis. The diagnosis of LVNC was confirmed by CMR in all patients according to Petersen’s criteria, with a median NC/C ratio of 3.09 (IQR: 2.46–3.73). Six patients were disqualified from CMR because of their severe clinical condition (*n* = 2), an implanted pacemaker (*n* = 2) and contraindications for anaesthesia (*n* = 2).

Symptoms of HF were present in ten patients (32%), including decreased LVEF in all ten patients and elevated NTproBNP values (normal value: up to 320 pg/ml) in five patients (16%). Arrhythmias and atrioventricular conduction disorders were observed in 15 children (48%). In this subgroup, the most common were sinus bradycardia (*n* = 9), paroxysmal second- or third-degree atrioventricular (AV) block (*n* = 4), episodes of non-sustained ventricular tachycardia (nsVT) (*n* = 2) and supraventricular tachycardia (SVT) (*n* = 2). One child underwent RF ablation. Electrocardiographic features of Wolff–Parkinson–White syndrome (WPW) were found in two children, one of whom underwent an electrophysiological (EPS) study in which no additional atrioventricular conduction was found. The second patient awaits an EPS study. Permanent pacemakers were implanted in two patients due to complete AV blockage (*n* = 1) and symptomatic sinus bradycardia (*n* = 1). Thromboembolic events occurred in two patients (6%). Mechanical circulatory support (LVAD) was implanted in a patient with extremely low LV systolic function; he was qualified for heart transplant, but died while waiting for a heart transplant. In the entire study group, deaths occurred in two children (6%).

Detailed clinical characteristics of the study group are summarised in Table 1. 

### 3.2. Family History

A positive family history of cardiomyopathy, arrhythmias, thromboembolic episodes and sudden cardiac death was found in 14 families (48%), being more common in those identified with the putative disease-causing variant than in those without it (Figure 1). 

As many as 12 children (including the 2 sets of siblings) had first-degree family members who were affected with cardiomyopathy—LVNC (*n* = 8), LVNC and HCM (*n* = 1), LVNC and DCM (*n* = 2) and HCM (*n* = 1). The arrhythmias observed in these families included bradycardia (*n* = 5), SVT (*n* = 1), atrioventricular node re-entry tachycardia (AVNRT) (*n* = 1) and WPW syndrome (*n* = 2). Sudden cardiac death (SCD) was noted in three families, including the 32-year-old father of patient P11, diagnosed with HCM, and the 1-month-old baby of P20’s aunt (postmortem examination was not performed). The third SCD occurred in the 2-year-old sister of P3, who had been diagnosed with LVNC and is described as P2 in the study.

### 3.3. Molecular Characteristics 

NGS was performed with a mean depth of 118×; the mean 20-fold coverage of target was 96%. Genotyping using a targeted cardiomyopathy-associated panel combined with Sanger analysis resulted in the identification of 16 unique variants in 11 genes in 16 patients (Table 2, Appendix A), yielding a 52% detection rate (15/29 families). Subsequent WES performed in two children who were unsolved in CMHI NGS 1000 panel analysis did not indicate a molecular diagnosis of LVNC. None of the affected patients in our cohort had a complex genotype with more than one gene involved.

Thirteen pathogenic or likely pathogenic variants in genes previously associated with LVNC aetiology, including variants detected in *ACTN2*, *HCCS*, *HCN4*, *LAMA4*, *MYH6*, *MYH7*, *PRDM16*, *TAFAZZIN* and *TTN*—as well as three rare variants of uncertain significance in *ACTC1* and *RBM20* genes were identified. There were 10 novel variants among them. Missense variants accounted for the largest proportion of identified changes (*n* = 8), followed by nonsense (*n* = 3), splice-site variants (*n* = 3), frameshift indel (*n* = 1) and gross deletion (*n* = 1). When considering the inheritance pattern of selected variants, two of them were X-linked (XLR or XLD), while the remaining variants apart from one were autosomal dominant (AD). The complete segregation analysis (both parents) performed in nine families revealed that in three cases (33%) the variant occurred de novo in the proband, while in five cases (56%) it was inherited from the affected parent. In one patient (P12), two biallelic *RBM20* variants were suggestive of autosomal recessive (AR) inheritance. 

The most frequent defects in our cohort were identified in the *HCN4*-encoding ion-channel protein (*n* = 4), in sarcomere *MYH7* (*n* = 2) and in the regulatory gene *PRDM16* (*n* = 2). The remaining variants were detected only once in single families (Figure 2). 

### 3.4. Characteristics of Patients with Putative Disease-Causing Variants

Among the sixteen patients (fifteen families) with confirmed molecular defects, the severe clinical outcome resulted in early death in two children. Patient P2 was diagnosed with LVNC, which presented with early HF symptoms and one thromboembolic event. She died at the age of 2 years due to the mechanism of sinus bradycardia related to the known pathogenic missense variant c.1444G>A p.(Gly482Arg) in the *HCN4* gene [14]. The second patient (P10), with the novel CNV variant c.(460+1_461-1)_(699+1_700-1)del p.? (a large deletion spanning exons 6–9) of the *TAFFAZIN* gene, was diagnosed with Barth syndrome with neutropenia and facial dysmorphic features. He presented symptoms of severe HF (NYHA class IV) with a significantly elevated NTproBNP value, an extremely reduced LVEF and a history of nsVT. He had an implanted left ventricular assist device and died waiting for a heart transplant. Adverse events were also noted in five patients, including two with HF and three individuals with a thromboembolic event, nsVT and pacemaker implantation, respectively. Among those who developed symptoms of HF (LVEF reduction and LV enlargement) were patients P9 (novel pathogenic *PRDM16* complex rearrangement c.1286_1289delinsTTGCACTT p.(Gly429Valfs*176)) and P7 with additional sinus bradycardia (novel likely pathogenic *MYH7* c.3973-2A>C p.? variant). A symptomatic sinus bradycardia was an indication for pacemaker implantation in P6 (novel *MYH7* pathogenic c.323G>A p.(Arg108His) variant). A thromboembolic event, without increasing symptoms of HF, was noted in P8 (novel *PRDM16* pathogenic c.1336G>T p.(Glu446*) variant), while nsVT was documented in P15 (novel *TTN* likely pathogenic c.44281+1G>T p.? variant).

The second syndromic LVNC in our cohort, in addition to Barth syndrome, was identified in P14, a girl with congenital microphthalmia resulting from the nonsense *HCCS* c.789G>A p.(Trp263*) molecular variant. She had an electrocardiographic pattern of WPW and episodes of SVT, thus she underwent an EPS study, which did not confirm the presence of an additional conduction pathway.

Patients with and without molecular defects presented with similar clinical and ECHO/CMR characteristics. The only specific phenotype related to a particular gene dysfunction was observed in four patients (P2–P5), who presented with LVNC accompanied by sinus bradycardia and the dilation of the ascending aorta resulting from known pathogenic *HCN4* variants. In one of the patients (P5), LGE was found (4.7% of the total LV mass). 

LV enlargement was observed only in one patient (P13) with an *ACTN2* molecular variant during ECG and CMR examinations, while the remaining patients with LP variant in *LAMA4* (P1) and with VUS in the *MYH6*, *RBM20* and *ACTC1* (P11, P12 and P16) presented no HF features. 

### 3.5. Patients with Unknown Molecular Etiology

No disease-causing variants were found in fifteen patients (14 families), including four children (P17, P18, P28 and P31) who showed symptoms of HF with decreased LVEF and LV enlargement, two (P21 and P22) with reduced LVEF only and one patient (P29) who demonstrated only LV enlargement in ECG and CMR studies. In this group, LGE (ranging from 3% to 9.5% of the total LV mass) was found in five patients (P21, P23, P29, P30 and P31). Among the three patients with sinus bradycardia (P29 through P31), two siblings (P30 and P31) with sick sinus syndrome demonstrated LVNC with dilatation of the ascending aorta, as did patients P2–P5, with *HCN4* alterations [14]. These siblings had paroxysmal second- or third-degree AV blockages, were receiving salbutamol therapy and were waiting for further cardiologic examinations in order to qualify for the implantation of permanent cardiac pacing. Patient P28, with complete AV blockage, had a pacemaker implanted, while P23, with SVT, had undergone RF ablation. The electrocardiographic pattern of WPW was noticed in P20, who was still awaiting EPS and RF ablation. 

Finally, we found that patients with and without molecular defects presented with distinct clinical and ECHO/CMR characteristics (Figure 1). While arrhythmias (mainly sinus bradycardia and nsVT), thromboembolic events and death were predominately observed in the group with molecular defects, the symptoms of HF and LGE were mainly found in the group without them.

## 4. Discussion

This study is the first large prospective study to report the molecular characteristics in Polish LVNC paediatric patients diagnosed in a single institute, which contributes to the still incomplete picture of disease aetiology in this age group. Paediatric LVNC has been reported with structural congenital heart diseases (CHD). Ventricular septal defect (VSD), atrium septum defect (ASD), persistent ductus arteriosus (PDA) and morbus Ebstein are the most reported forms of CHD in association with LVNC in the paediatric population. In our group, we analysed patients without congenital heart disease, which was the exclusion criterion for the study.

Though there is a general consensus on the importance of genetic testing, the appropriate targets of testing are still a matter of debate [24,25]. The Heart Rhythm Society and the European Heart Rhythm Association guidelines on genetic testing for CMPs state that molecular testing can be useful in any case suspected for LVNC (class IIa recommendation) [26]. Some emphasise that genetic screening is mostly beneficial in LVNC that is associated with other cardiac and syndromic features, in which case it can facilitate the proper diagnosis, while it is less useful in cases with isolated LVNC without a family history [25]. The authors of the current paper and other researchers demonstrated that DNA testing should not be restricted to only cases with a positive family history, as the detection of the causative molecular variant can help in the accurate identification of carriers with an increased risk of adverse events and may guide proper clinical management. On the other hand, this can be beneficial for non-carrier relatives who may be excluded from regular cardiac follow-up and who can be reassured that their offspring demonstrate no increased risk [24]. In this study, a positive family history of cardiomyopathy, arrhythmias, thromboembolic episodes and sudden cardiac death was noted in almost half of the LVNC group; this was one of the main reasons for referring the children for cardiologic diagnosis. In the majority of these families, the molecular basis of the LVNC was confirmed (*n* = 9). 

The development of NGS technologies has resulted in a broader molecular spectrum related to LVNC. However, the data are still limited and inconsistent with regard to molecular heterogeneity and diagnostic yield in age-diverse cohorts, especially in children. This can be due to the rarity of childhood LVNC, the lack of widespread routine genetic testing and different gene panels being used. The strategies for designing screening panels varies widely, as one method includes only genes with a proven LVNC association, while others also test for genes potentially related to a broad spectrum of various cardiomyopathies [27]. In our study, we used a large panel of more than 200 CMP-associated genes to reduce the possibility of overlooking potential deleterious variants, and this approach resulted in a high diagnostic rate (15/29 families; 52%). Notably, we also considered, as a positive result, VUSs that were identified in genes with clinical evidence of LVNC as being highly likely for pathogenic designation upon future co-segregation analysis and functional studies that would confirm its impact on the protein. The lack of molecular diagnosis in the remaining 48% of our group might be due to possible pitfalls in variant prioritisation or interpretation (e.g., deep intronic or synonymous variants altered splicing missed), the occurrence of variants in difficult-to-detect regions (e.g., GC-rich regions or homopolymeric repeats) or the limited number of genes in the selected panel. Therefore, in cases with no variants of interest found by deep targeted sequencing, future studies should include whole exome or genome sequencing. However, WES analysis performed in two patients in our group did not provide a solution. 

Several studies applying NGS in LVNC testing demonstrated various diagnostic rates and genetic spectra in age-diverse cohorts, while others reported similar results regardless of whether children or adults were studied [28]. Van Waning et al. [29] reviewed papers on the clinical and molecular screening of LVNC cases that were published between January 1999 and March 2018. This collection (*n* = 561) contained 244 children, in whom 195 unique variants were identified. The more prevalent changes were related to *MYH7*, *MYBPC3*, *ACTC1*, *TTN*, *TAFAZZIN* and *HCN4* dysfunction. They observed that syndromic, mitochondrial and chromosomal defects were frequently detected in paediatric patients with severe outcome. Within the largest single-centre paediatric LVNC cohort described so far (*n* = 206, age 0–16 years), Hirono et al. [30] confirmed molecular aetiology in 87 patients. Since they used a wide panel of 182 cardiac disorder-related genes, 99 pathogenic variants were detected in 40 genes, with the most frequently altered ones being *MYH7*, *TAFAZZIN* and *ANK2*, followed by *TPM1*, *SCO2*, *ACTC1*, *KCNQ1*, *MYBPC3*, *MYL2*, *ERBB2* and *HCN4*. The heterogeneous molecular spectrum reflected the diverse study group, comprising patients with LVNC-DCM, LVNC arrhythmia, and LVNC congenital heart disease as well as those with isolated LVNC. The reported diagnostic rate in the isolated LVNC subgroup (*n* = 38) was 50%. Interestingly, one third of Hirono’s whole group had arrhythmia. Even more frequent arrhythmias were observed in our patients, mostly in those with likely disease-causing variants. Miller et al. [26] also described a large cohort of 151 patients under the age of 21 years, including 61 patients diagnosed with isolated LVNC who were molecularly tested using 11–38 gene panels. In contrast to the above reports, pathogenic and likely pathogenic variants were detected in 9% of the whole group (mostly *MYH7*, *MYBPC3*, *TPM1* and *TNNT2* variants); however, none of the isolated LVNC patients had a positive result. The authors pointed to differences in clinical diagnostics of LVNC (isolated/cardiomyopathy-associated), various panel content or the interpretation of the pathogenicity of the identified variants (they did not include VUSs) as potential explanations of the discrepancies they observed. 

In line with studies concerning childhood and adult LVNC cases, we also found that *MYH7* appeared to be a significant cause of disease, since pathogenic/likely pathogenic variants in this gene account for 13% of all identified alterations within our group. As it was observed that variants in this gene were independent risk factors for adverse events [1], our patients with *MYH7* variants were among the severely affected cases, both of whom presented with sinus bradycardia. In one case (P6), this led to peacemaker implantation, while in the second case (P7) the disease course was complicated by HF. Recently, *MYH7* pathogenic variants and VUSs have also been frequently identified in foetal-onset LVNC [31]. 

The *HCN4* variants were the most commonly identified in our study (20%; P2–P5) which is in line with the increasing number of reports confirming a significant contribution of the *HCN4* gene in LVNC aetiology in both children and adults [29,30,32,33,34]. This resulted in the recent inclusion of *HCN4* into commonly used CMP-related gene panels. *HCN4* encodes the hyperpolarisation-activated cyclic nucleotide-gated channel 4, which plays a crucial role in proper pacemaker activity and conduction system functioning. Pathogenic *HCN4* alterations have been associated with a broad spectrum of conditions, mainly sick sinus syndrome and Brugada syndrome, but also with LVNC, sinus bradycardia, sinus tachycardia, atrial fibrillation, atrioventricular block, idiopathic ventricular tachycardia, myocardial infraction, sudden infant death syndrome, ARVC, dilation of the aorta and chronotropic incompetence [35,36]. Notably, in our patients the *HCN4* pathogenic variants were associated with specific mixed phenotypes of LVNC, sinus bradycardia and substantial dilation of the ascending aorta [14]. We observed an early onset and fatal course with progressive HF complicated by an embolic event that resulted in childhood death in patient P2. 

Our study revealed two further pathogenic variants in the *PRDM16* gene. PRDM16 functions as a compact myocardium-enriched transcription factor and is involved in the activation of genes required for compact myocardium growth and in the repression of genes associated with trabeculae formation. In a recently published mice model study, it was suggested that the development of LVNC could result from a shift in the transcriptional profile of compact cardiomyocytes [9]. In a large meta-analysis of LVNC cases, Mazarotto et al. [27] showed truncating variants in *PRDM16* to be significantly enriched in severe LVNC cases as compared with other CMPs. Both patients within our group experienced adverse events: HF (P9) and an embolic event (P8). 

Sarcomere genes are implicated as genetic triggers in the development of LVNC, regulating the expression of numerous genes involved in heart development or modifying the severity of disease [37]. We identified putative disease-causing variants in a significant proportion (40%) of cases, similar to a finding observed earlier [1,29,30]. Apart from the most common *MYH7* gene, *TTN* seems to be frequently reported in LVNC patients, including children [27], and is often associated with a higher risk of LV systolic dysfunction and adverse events [38]. The only presumably pathogenic *TTN* allele found in this study in P15 was inherited from his affected father. This variant c.44281+1G>T altered correct splicing but functional studies are required to determine its impact on the protein. The disease course presented in P15 was severe with significant cardiac arrhythmia and episodes of nsVT, but without HF features. Other variants enriched in LVNC are *ACTN2* truncating variants [27], consistent with our finding in P2, with relatively mild clinical course associated with LV enlargement only, without progressive HF. An *in silico* analysis showed a likely pathogenic nature of the identified variant, but the lack of available family members prevented an assessment of variant segregation. In two other mildly affected patients, missense variants in sarcomeric genes were identified, including *ACTC1* (c.329C>T p.(Ala110Val)) and *MYH6* (c.4850A>C p.(Lys1617Thr)). They were classified as VUSs, but might still be disease-causing variants if pathogenicity is confirmed with cosegregation among family members. This was not possible in this study due to a lack of contact with the parents or their refusal to participate. 

Surprisingly, in patient P12 we identified biallelic variants that altered the *RBM20* gene, which encodes RNA-binding motif protein 20, a relevant splice regulator of sarcomeric and calcium-handling genes important for cardiac functioning [39]. Since *RBM20*-related CMP manifests mainly in DCM patients (rarely in HCM, ARVC or LVNC) and is correlated with high rates of HF, arrhythmias and sudden cardiac death [40], the relatively mild clinical course observed in our patient was not consistent with previous reports. Notably, 2 rare, possibly deleterious variants are suggestive of autosomal recessive inheritance, while all cases with *RBM20* alterations described so far have been autosomal dominant. Further functional studies are critical in order to validate our findings.

Contrary to other studies demonstrating *TAFAZZIN* dysfunction among the most common paediatric LVNC causes [1], only one such patient was detected within our study group. It is likely that more Barth patients were diagnosed in the CMHI Department of Paediatrics, Nutrition and Metabolic Diseases because of the specific clinical picture of the metabolic disorder, i.e., the pronounced 3-methylglutaconate aciduria. According to the literature on the subject [41], one of our patients demonstrated one of the most severe clinical courses, with progressive HF and life-threatening arrhythmias such as nsVT. Despite pharmacological treatment and LVAD implantation, he died awaiting heart transplantation.

In contrast to previous reports, we did not detect LP/P variants in *MYBPC3* or the multiple variants commonly reported in LVNC, a fact which could be associated with the relatively small sample. 

It is worth noting that we could not confirm the reports from other authors regarding LV systolic dysfunction and a worse prognosis in patients with an established molecular diagnosis [1,24], since within our group symptoms of HF were predominant in the patients with no pathogenic molecular variant identified. Pathogenic molecular variants were more prevalent among the female patients (67% of the girls vs. 31% of the boys). In our study, thromboembolic events only occurred in the girls, but we found no correlation between the patient’s gender and the NC/C ratio or the presence of HF, arrhythmias or sudden cardiac death.

Although genetic tests are not used in the routine diagnosis of LVNC patients, they demonstrate increasing value and their results may have an impact on the clinical course of the disease, prognosis and medical care of the patient and family members.

## 5. Study Limitations

One limitation of this work was the small study group, preventing us from making stronger conclusions on genotype–phenotype correlations and prognosis. International collaboration in creating a collective data registry would pave the way for the better sharing of molecular and clinical data to establish gene–disease associations. 

Additionally, we were unable to follow all patients for a long period of time, especially those patients who were referred from external centres. Since we used an NGS panel targeted at known CMP-associated genes, variants in novel genes were likely to have been missed, especially as a gene coverage analysis of major cardiomyopathy genes was not performed in our study. We cannot exclude the possibility that a subset of identified VUSs may also be reclassified as likely pathogenic or pathogenic variants with further available data on segregation and/or functional investigations, which was not performed in this study. Some parental samples were unavailable for segregation analysis, which reduced the ability to determine inheritance patterns or to predict genetic recurrence in these families. 

## 6. Conclusions

Our study confirmed a highly divergent molecular spectrum of LVNC in children. The results of our research demonstrate that the comprehensive testing of a wide panel of CMP-related genes helped to explain the molecular aetiology in more than half of the group of patients. The molecular defect in most children correlated with the occurrence of arrhythmias, death and a family history of CMP. Genetic studies helped to identify cases with specific cardiac phenotypes that are prone to severe outcome and adverse events, while a negative genetic test result did not exclude the clinical recognition of LVNC. Genetic counselling is an integral part of work-up and management of LVNC in children.

## Figures and Tables

**Figure 1 genes-13-01334-f001:**
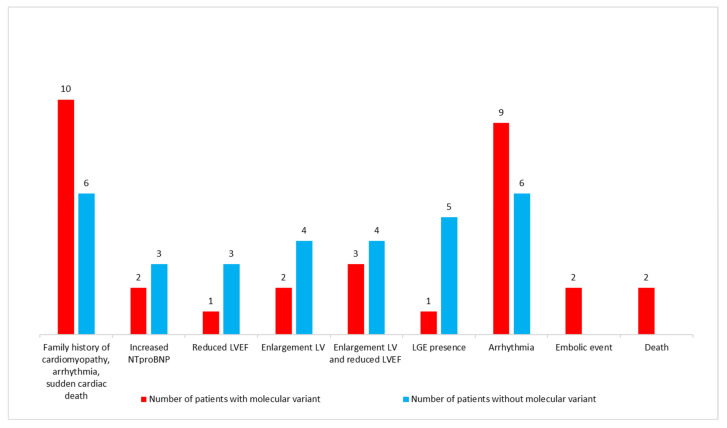
Comparison of clinical data in patients with/without molecular variants.

**Figure 2 genes-13-01334-f002:**
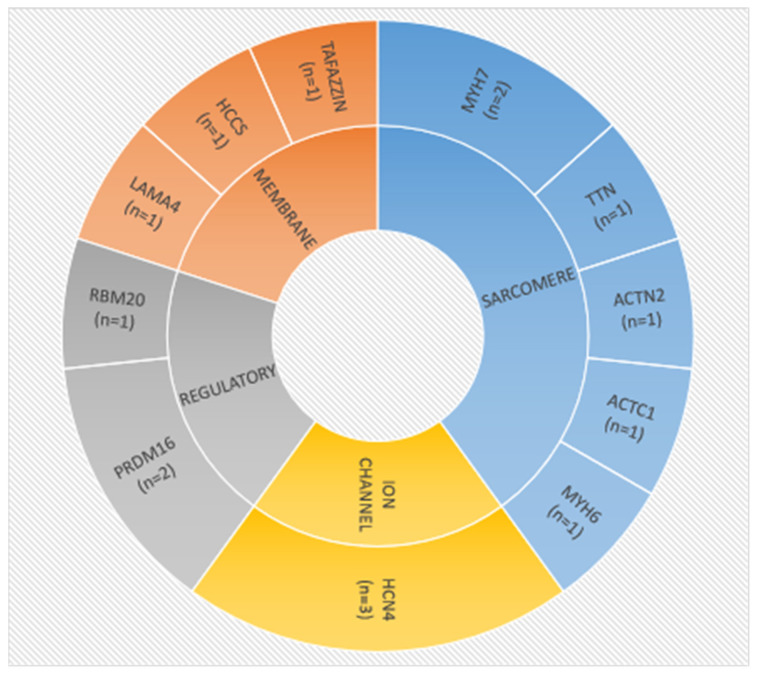
Distribution of LVNC-related genes identified in this study. The number of families carrying putative disease-causing molecular variants in particular genes are in brackets. The inner circle indicates the category of proteins involved.

**Table 1 genes-13-01334-t001:** Clinical characteristics of the patients.

Patient	Age (Yers)	Gender	Syncope	Embolic Event	Dysmorphic Features	Heart Failure Profile						Arrhythmias and Atrioventricular Conduction Disorders	EPS/RFA	Pacemaker	LVAD	Death	Family History			NC/C (Echo)	NC/C (CMR)	LGE %	Affected Gene
						NYHA/Ross Class	NTproBNP pg/mL	LVEF (ECHO)	LVDd mm (z-Score) ECHO	LVEF (CMR)	LVEDV mL/BSA (z-Score)						CMP	Arrhythmias	SCD				
P1	1	F	0	0	0	II	256.50	63	32.2 (2)	N/A	N/A	0	0	0	0	0	0	0	0	6.3	N/A	N/A	*LAMA4*
P2	2	F	0	1	0	IV	N/A	59	28.4 (0.5)	N/A	N/A	Brady	0	0	0	1	LVNC (sister)	Brady (sister, father’s father)	0	3.34	N/A	N/A	*HCN4*
P3	6	F	0	0	0	II	110.80	64	35.6 (−0.5)	71	73 (0.7)	Brady	0	0	0	0	LVNC (sister)	Brady (sister, father’s father)	1	2.35	4.6	0	*HCN4*
P4	11	F	0	0	0	I	<5.0	70	46.3 (0.9)	71.3	69.6 (1.2)	Brady	0	0	0	0	LVNC (mother, 2 cousins)	0	0	2.22	3.0	0	*HCN4*
P5	17	M	0	0	0	II	32.95	71.5	41.1 (0.8)	63.7	125.8 (−1)	Brady	0	0	0	0	LVNC (sister), HCM (father)	Brady (2 sisters)	0	2.4	3.73	4.7	*HCN4*
P6	5	F	1	0	0	II	N/A	61	38 (−0.1)	N/A	N/A	Brady	0	1	0	0	0	0	0	2.85	N/A	N/A	*MYH7*
P7	12	F	1	0	0	II	68.54	50	52.2 (2.1)	54.5	103.5 (3)	Brady	0	0	0	0	LVNC (mother)	0	0	2.6	3.38	0	*MYH7*
P8	6	F	0	1	0	II	38.69	61	37 (0.4)	56	71 (0.9)	0	0	0	0	0	0	0	0	2.26	2.3	0	*PRDM16*
P9	11	F	0	0	0	II	331.20	52	54.2 (3.6)	52.4	80.9 (1.4)	0	0	0	0	0	0	0	0	2.6	3.52	0	*PRDM16*
P10	6	M	0	0	1	IV	27057.0	28	36 (3.5)	N/A	N/A	nsVT	0	0	1	1	0	0	0	3.9	N/A	N/A	*TAFFAZIN*
P11	6	F	0	0	0	I	99.89	64	37.3 (−0.4)	68	68.6 (0.9)	0	0	0	0	0	HCM (father, father’s sister and father’s mother)	0	1	2.7	3.09	0	*MYH6*
P12	11	F	0	0	0	II	62.91	56	34 (0.3)	68.7	66.7 (−3.6)	0	0	0	0	0	0	SVT (father’s father)	0	4.4	3.76	0	*RBM20*
P13	13	F	0	0	0	II	33.30	57	57 (3.6)	55.9	90.9 (1.8)	0	0	0	0	0	0	0	0	3.9	5.74	0	*ACTN2*
P14	15	F	0	0	1	II	13.32	63	45 (−0.1)	74.2	77 (0.5)	WPW, SVT	1	0	0	0	0	WPW (father’s father)	0	2.22	2.46	0	*HCCS*
P15	16	M	0	0	0	II	131.70	59	52.3 (3)	55.6	79.5 (−1.5)	nsVT	0	0	0	0	LVNC (father)	0	0	2.13	2.43	0	*TTN*
P16	17	M	0	0	0	II	32.73	70	52.6 (−0.2)	62.2	78.8 (1.1)	0	0	0	0	0	LVNC (mother, sister, mother’s father)	0	0	3.3	2.26	0	*ACTC1*
P17	0.2	F	0	0	0	II	577.80	54	23.5 (2.8)	N/A	N/A	0	0	0	0	0	LVNC (father)	0	0	4.8	N/A	N/A	0
P18	5	F	0	0	0	II	1325.0	50	42 (2.5)	61.6	76.3 (1.2)	0	0	0	0	0	0	0	0	4.0	4.22	0	0
P19	5	M	0	0	0	II	65.34	63	36.4 (0.8)	53	78 (1.1)	0	0	0	0	0	0	0	0	3.4	2.65	0	0
P20	6	M	0	0	0	II	98.64	57	37.1 (1)	67.6	104.3 (2.1)	WPW	0	0	0	0	0	WPW (father’s sister)	1	2.32	2.36	0	0
P21	6	M	0	0	0	II	118.80	53	36.6 (0.8)	54.2	54.4 (0)	0	0	0	0	0	0	0	0	2.8	2.63	9.5	0
P22	8	M	0	0	0	II	349.40	55	37.8 (−0.5)	48.1	56.5 (−0.7)	0	0	0	0	0	LVNC (mother)	0	0	2.13	2.39	0	0
P23	10	F	0	0	0	II	63.01	51	32.8 (−1.7)	71.7	61.1 (−1.8)	SVT	1	0	0	0	0	0	0	2.2	3.27	7.8	0
P24	11	M	0	0	0	II	38.88	63	56 (4.6)	63	88.7 (0.9)	0	0	0	0	0	0	0	0	5.14	5.07	0	0
P25	13	F	0	0	0	II	67.14	56	51 (2.6)	58.4	83.2 (0.7)	0	0	0	0	0	0	0	0	2.22	3.45	0	0
P26	13	M	0	0	0	I	81.78	64	51.5 (1.9)	62.9	85.8 (0.9)	0	0	0	0	0	0	0	0	2.9	2.60	0	0
P27	13	F	1	0	0	II	30.07	60	35.7 (−2)	72.9	60.2 (−1.1)	0	0	0	0	0	0	AVNRT (mother)	0	2.85	2.56	0	0
P28	15	M	0	0	0	II	73.13	53	51.8 (2.8)	N/A	N/A	AV block	0	1	0	0	0	0	0	3.1	N/A	0	0
P29	16	M	0	0	0	II	8.76	65	58 (2.8)	57.3	130 (3)	Brady, AV block	0	0	0	0	0	0	0	3.9	2.45	3	0
P30	11	M	0	0	0	II	19.04	62	47.3 (2.3)	70.2	114.1 (1.2)	Brady, AV block	0	0	0	0	LVNC (sister); DCM (father)	Brady (sister, father, father’s father)	0	2.13	3.27	8	0
P31	16	F	0	0	0	II	91.10	55	59.5 (2.9)	54.2	122.1 (3.7)	Brady, AV block	0	0	0	0	LVNC (brother); DCM (father)	Brady (brother, father, father’s father)	0	2.06	3.91	6.7	0

P1–P31—Patient numbers; AV block—paroxysmal second-/third-degree atrioventricular block; AVNRT—atrioventricular node re-entry tachycardia; Brady—sinus bradycardia; BSA – body surface area; CMP—cardiomyopathy; CMR—cardiovascular magnetic resonance; DCM—dilated cardiomyopathy; EPS—electrophysiology study; HCM—hypertrophic cardiomyopathy; LGE—late gadolinium enhancement; LVAD—left ventricular assist device; LVDd—left ventricular diastolic diameter; LVEDV—left ventricular end-diastolic volume; LVEF—left ventricular ejection fraction; LVNC—left ventricular noncompaction cardiomyopathy; NC/C—noncompacted to compacted myocardial layer ratio; nsVT—non-sustained ventricular tachycardia; NTproBNP—N-terminal pro-brain natriuretic peptide; NYHA—New York Heart Association class; RFA—radiofrequency ablation; SCD – sudden cardiac death; SVT—supraventricular tachycardia; WPW—Wolff–Parkinson–White syndrome; yrs—years; N/A—data are not available; ‘0’—not present; ‘1’—present.

**Table 2 genes-13-01334-t002:** List of putative disease-causing variants identified in this study.

Gene	Chromosome	Transcript	Nucleotide Change	Protein Effect	Variant Type (Consequence)	Pathogenicity	Patient	Family Segregation	CMP Family History	References
*ACTC1*	15	NM_005159.5	c.329C>T	p.(Ala110Val)	missense	VUS	P16	NA	LVNC (mother, sister, mother’s father)	[5]
*ACTN2*	1	NM_001103.4	c.1163G>A	p.(Trp388*)	nonsense (LOF)	P	P13	NA	no	this study
*HCCS*	X	NM_005333.5	c.789G>A	p.(Trp263*)	nonsense (LOF)	P	P14	NA	no	this study
*HCN4*	15	NM_005477.3	c.1444G>A	p.(Gly482Arg)	missense	P	P2, P3 (siblings)	paternal	LVNC (sister)	[14]
*HCN4*	15	NM_005477.3	c.1454C>T	p.(Ala485Val)	missense	P	P4	maternal	LVNC (mother, 2 cousins)	[14]
*HCN4*	15	NM_005477.3	c.1438G>C	p.(Gly480Arg)	missense	LP	P5	paternal	LVNC (sister); HCM (father)	[14]
*LAMA4*	6	NM_002290.5	c.719-1G>T	p.?	splicing (LOF)	LP	P1	NA	no	this study
*MYH6*	14	NM_002471.4	c.4850A>C	p.(Lys1617Thr)	missense	VUS	P11	NA	HCM (father, father’s sister and father’s mother)	[5]
*MYH7*	14	NM_000257.4	c.323G>A	p.(Arg108His)	missense	P	P6	de novo	no	this study
*MYH7*	14	NM_000257.4	c.3973-2A>C	p.?	splicing (LOF)	P	P7	maternal	LVNC (mother)	this study
*PRDM16*	1	NM_022114.4	c.1336G>T	p.(Glu446*)	nonsense (LOF)	P	P8	de novo	no	this study
*PRDM16*	1	NM_022114.4	c.1286_1289delinsTTGCACTT	p.(Gly429Valfs*176)	indel (LOF)	P	P9	de novo	no	this study
*RBM20*	10	NM_001134363.3	c.1232C>T	p.(Pro411Leu)	missense	VUS	P12	maternal	no	this study
*RBM20*	10	NM_001134363.3	c.1958C>T	p.(Thr653Ile)	missense	VUS	P12	paternal	no	[23]
*TAFAZZIN*	X	NM_000116.5	c.(460+1_461-1)_(699+1_700-1)del	p.?	gross deletion (LOF)	P	P10	NA	no	this study
*TTN*	2	NM_001267550.2	c.44281+1G>T	p.?	splicing (LOF)	LP	P15	paternal	LVNC (father)	this study

CMP—cardiomyopathy; HCM—hypertrophic cardiomyopathy; LVNC—left ventricular noncompaction cardiomyopathy; LOF—loss of function; LP—likely pathogenic variant; P—pathogenic variant; VUS—variant of uncertain significance; NA—not analysed. The nomenclature of molecular variants follows the Human Genome Variation Society’s guidelines (HGVS; http://varnomen.hgvs.org/), using human cDNA reference sequences following the Human Gene Mutation Database (HGMD; http://www.hgmd.cf.ac.uk).

## Data Availability

The data presented in this study are available on request from the corresponding author.

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
