# Peer review of "Genetic Profile of Left Ventricular Noncompaction Cardiomyopathy in Children—A Single Reference Center Experience"

_genes, 2022, doi:10.3390/genes13081334_

Round 1
Reviewer 1 Report
Review Comments
Summary
LVNC may be an isolated clinical phenotype or be associated with a genetic cardiomyopathy and an increased risk for heart failure, atrial and ventricular arrhythmia and thromboembolic stroke. Previous studies in the adult population have suggested that 20-40% of LVNC may be familial. Based on prior estimated that 10-50% of patient may have an underlying genetic cause, as shown by genetic mutations in MYH7, TTN, MYBPC3 and others. The prevalence of genetic mutation among pediatric patients with isolated LVNC is unknown. The authors performed targeted gene panel sequencing and identified 16 distinct genetic variants in 11 genes among 31 pediatric patients with LVNC by echo or CMR. Clinical features of these children were discussed in the context of their family history and genes affected. Of note, a molecular diagnosis was made in 52% of the children but it was not associated with increased rates of heart failure in this cohort. The high prevalence of family history and genetic diagnosis argues for genetic evaluation among children with LVNC.
Abstract
- The word ‘genetic background’ typically refers to polygenic risk rather than monogenic mutation assessed in this study. I would suggest that the authors rephrase the section under methods to describe their method more precisely.
Methods
- The process of variant classification should be described in further detail as the authors stated that 10 variants were novel (not previously observed or reported in ClinVar for instance). Please use standard nomenclature of pathogenic and likely pathogenic to describe genetic variants identified and describe methods by which variant classification was conducted.
Result
- What is the definition of increased LV EF?
- It is somewhat surprising that HCN4 was the most common gene with mutations found. In the methods, it is stated that 31 patients from 29 families (including 2 sets of siblings) hospitalized between 2008 and 2021 were included. Why were these children hospitalized? Were most children hospitalized for ablation or arrhythmia treatment?
Discussion
- Whole exome sequencing in most cases is unlikely to lead to variant classification to pathogenic due to insufficient evidence for phenotype-gene association. It is not surprising that whole exome sequencing did not yield additional results unless de novo variants were identified.
- Please discuss the role of confounding factors such as hospitalization or presenting diagnosis that may bias the cohort towards less heart failure or having a higher proportion of children with HCN4 compared to prior larger cohorts.
Author Response
Warsaw, June 17, 2022
RE: Manuscript ID: genes-1716401
Title: Genetic profile of left ventricular noncompaction cardiomyopathy in children: A single reference centre experience
Dear Prof. Dr. J. Peter W. Young,
Dear Reviewers,
Thank you very much for your response and for submitting our manuscript for revision. Thank you to the Reviewers for a thorough and very helpful analysis of our research work and for all comments, suggestions and questions.
We have reviewed the manuscript and performed the suggested improvements (the 'track changes' function was used to mark the changes). Our answers to the Reviewers questions and comments as well as the resulting changes are described in detail below. The sentences added to the main manuscript are highlighted in red (‘track changes’).

Reviewer 2 Report
The current study attempts to report the genetic screening of LVNC in a group of children examined in a single pediatric reference center. The study is interesting and provides valuable information on clinical and genetic characteristics of LVNC in the pediatrics population. However, this reviewer has some concerns for the present version of the manuscript. The text needs English language correction prior to be publication ready.
Comments,
1. I have few suggestions for methods section and data presentation in the results which will make the manuscript easy to follow for readers. Please explain results under subheadings, ex. “Patients selection & Data collection”; “Ethics Statement”; “Gene panel design”;” Next generation sequencing and data analysis”; “Statistical Analysis” if any.
2. Please add exclusion criteria for patients’ selection. Did the authors consider secondary cardiomyopathies in their exclusion criteria?
3. In NGS section, please add a Variant filtration workflow
4. This reviewer feels this unique dataset in the result section is not explored enough.
5. Did the authors notice any correlation between degree of noncompaction involvement and sequencing results?
6. Did the authors consider any concomitant cardiac defects and their correlation to NGS results?
7. L167 Please explain “severe clinical condition”
8. Page 9, L20. Please add details for “52% detection rate”. What is the % coverage of investigated exons in NGS ?
9. For table 2, Please add a column for mutation type
10. Considering the small sample size, limitations for the study needs to be explained in the discussion. Besides dysmorphic features such as right ventricular or biventricular noncompaction may be explored as shortcomings as well.
There are multiple typos, grammatical errors, incomplete sentences throughput the manuscript (few of these are Page 1, Line 16, 34; Page 14, L88).
Author Response
Warsaw, June 17, 2022
Manuscript ID: genes-1716401
Title: Genetic profile of left ventricular noncompaction cardiomyopathy in children: A single reference centre experience
Dear Prof. Dr. J. Peter W. Young,
Dear Reviewers,
Thank you very much for your response and for submitting our manuscript for revision. Thank you to the Reviewers for a thorough and very helpful analysis of our research work and for all comments, suggestions and questions.
We have reviewed the manuscript and performed the suggested improvements (the 'track changes' function was used to mark the changes). Our answers to the Reviewers questions and comments as well as the resulting changes are described in detail below. The sentences added to the main manuscript are highlighted in red (‘track changes’).

Reviewer 3 Report
This study would undoubtedly contribute to this age group's still incomplete picture of disease etiology. This study will also certainly expand the genetic and clinical spectrum of the childhood LVNC. Overall, the experiments were appropriately designed and performed. The results also supported the author's conclusions. To further improve the manuscript, here are some comments which need to be addressed by the authors.
Comment 1. The authors should briefly discuss their study's limitations in the discussion section of their paper, as previously suggested by Rohde, Sofie, et al. in their review. "State-of-the-art review: Noncompaction cardiomyopathy in pediatric patients." Heart failure reviews vol. 27,1 (2022): 15-28. doi:10.1007/s10741-021-10089-7) They have suggested that an effort be made to collect cases in a multicenter, international registry to evaluate the outcome and impact of management strategies in these patients.
Comment 2. As discussed in previous findings, the pediatric NCCM has been reported with structural CHD. Such as Ventricular septal defect (VSD), atrium septum defect (ASD), persistent ductus arteriosus (PDA), and Morbus Ebstein are the most reported forms of CHD in association with NCCM in the pediatric populations, etc. The authors should briefly discuss these in their paper unless they are not relevant to their study. (Ozkutlu S, Ayabakan C, Celiker A, Elshershari H. Noncompaction of ventricular myocardium: a study of twelve patients. J Am Soc Echocardiogr. 2002 Dec;15(12):1523-8. doi: 10.1067/mje.2002.128212. PMID: 12464922 and Zuckerman WA, Richmond ME, Singh RK, Carroll SJ, Starc TJ, Addonizio LJ. Left ventricular noncompaction in a pediatric population: predictors of survival. Pediatr Cardiol. 2011 Apr;32(4):406-12. doi: 10.1007/s00246-010-9868-5. Epub 2010 Dec 25. PMID: 2118837
Comment 3. The study was carried out on eighteen female and thirteen male patients; however, there was no discussion about the gender specifics implications of their findings. Towards this, a table describing the different outcomes will help understand the consequences of their results in genders. Alternatively, the authors should discuss the gender-based results in their discussion. Are they significant?
Comment 4. Most importantly, the authors have not performed any statistical analysis throughout their study. Please clarify whether the analysis was not required in this study?
Author Response

(The authors gave the same response as above.)
